# Approximated Behavioral Metric-based State Projection for Federated Reinforcement Learning

## Abstract

Federated reinforcement learning (FRL) methods usually share the encrypted local state or policy information and help each client to learn from others while preserving everyone's privacy. In this work, we propose that sharing the approximated behavior metric-based state projection function is a promising way to enhance the performance of FRL and concurrently provides an effective protection of sensitive information. We introduce FedRAG, a FRL framework to learn a computationally practical projection function of states for each client and aggregating the parameters of projection functions at a central server. The FedRAG approach shares no sensitive task-specific information, yet provides information gain for each client. We conduct extensive experiments on the DeepMind Control Suite to demonstrate insightful results.

## 1 Introduction

In recent years, federated learning has emerged as a new approach to enable data owners to collaboratively train each one's improved local model with the help of the privacy preserved information from others (Yang et al., 2019a;b; Li et al., 2020a; Wei et al., 2020; Lyu et al., 2020). Federated reinforcement learning (FRL) applies federated learning principles to reinforcement learning (Zhuo et al., 2019). In FRL, multiple clients, each with their own local environments, collaborate to learn a collective optimal policy (Qi et al., 2021).

Aggregating knowledge from clients in non-identical environments allows FRL to explore a huge state-action space, enhance sample efficiency and accelerate the learning process (Wang et al., 2020). However, FRL faces unique challenges primarily due to the different local environments and diverse data distributions among clients. In FRL, clients may experience very different states and rewards in their own environment, resulting in diverse data distribution. This diversity may lead to significant differences in the learning model, making it difficult for clients to converge to a robust common policy (Zhao et al., 2018). Additionally, FRL must ensure that sensitive information remains protected from exposure to other clients or the central server (Zhu et al., 2019; Anwar & Raychowdhury, 2021).

Previous researches found that learning representation based behavioral metric can significantly accelerate the reinforcement learning process and enhance the generality of policy (Zhang et al., 2020; Agarwal et al., 2021; Kemertas & Aumentado-Armstrong, 2021). This method involves learning a state projection function by evaluating the behavioral similarities between states, which are measured in terms of rewards and state transition probabilities. The state projection function is valuable to the learning process, yet it does not reveal any sensitive task-specific information. In the FRL settings, clients would not directly share the rewards and state information because of the privacy issues. Therefore, sharing the parameters of the state projection function could be a promising research direction for FRL.

In this work, we propose the Federated Reinforcement Learning with Reducing Approximation Gap (FedRAG), a novel FRL framework to share parameters of state projection functions and to learn a local behavioral metric-based state projection function for each client. We detail FedRAG's network architecture in Figure 1, emphasizing how client collaboration is achieved through shared state projection functions. The global state projection function is formed by aggregating local state

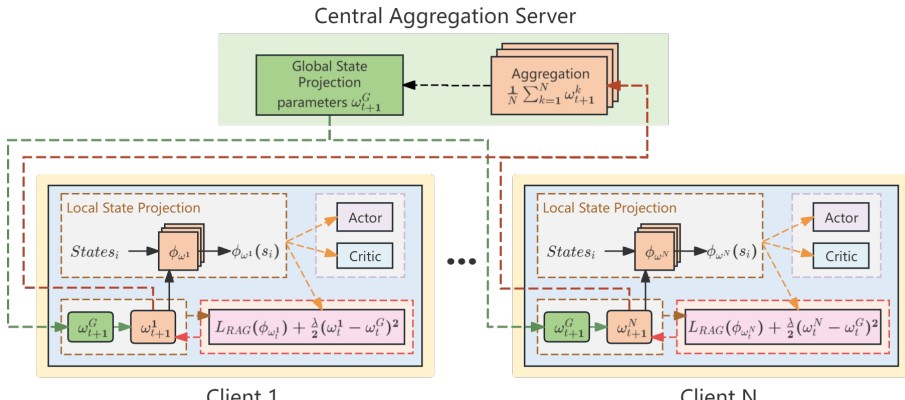

Figure 1: Framework of FedRAG. Periodically, the local state projection function parameters are synchronized to a central server. Then the central server distributes the averaged parameters to the clients. For each client, a regularization term is incorporated to ensure that the client's local state projection parameters follow the global updates.

projection functions, each trained with behavioral metrics to capture the unique transition dynamics and rewards of its respective environment. By integrating these locally learned features, the global state projection function reflects the diverse dynamics and rewards across different environments. Periodically, each client's local state projection function is replaced with the global state projection function, while the L2 regularization is continuously applied to maintain alignment throughout the learning process. Together, these mechanisms improve local state projection function and strategies that are robust and adaptable across varied environments. The main contributions are as follows:

- We propose FedRAG, a novel federated reinforcement learning framework to share the projection function of states, instead of traditionally sharing the encrypted states information. Subsequent analysis show that our method is beneficial to privacy-preserving as a side-effect.
- Under the FedRAG framework, we introduce a behavioral metric-based state projection function and develop its practical approximation algorithm in Federated Learning settings. Empirical results demonstrate our method is effective.

## 2 RELATED WORK

**Federated Learning**   Federated Learning (FL) was first introduced in FedAvg by McMahan et al. (2017), where training data remains distributed across mobile devices, and a shared model is learned by aggregating locally computed updates through iterative model averaging. Subsequently, FedProx, proposed by Li et al. (2020b), addresses system heterogeneity and statistical variability in federated networks. It incorporates a proximal term into local optimizations, allowing for variable computational efforts across devices, which helps stabilize diverse local updates. To accommodate the inherent heterogeneity in FL, Per-FedAvg, introduced by Fallah et al. (2020), was developed as a personalized approach. This method adapts Model-Agnostic Meta-Learning (MAML) to provide a suitable initial model that quickly adapts to each user's local data after training. Another innovation, pFedMe, proposed by T Dinh et al. (2020), tackles the statistical diversity among clients by utilizing Moreau envelopes as client-specific regularized loss functions, effectively decoupling personalized model optimization from global model learning.

**Federated Representation Learning**   Recently, federated representation learning, which focuses on training models to extract effective feature representations directly from raw data, has become increasingly popular. LG-FedAvg, proposed by Liang et al. (2020), optimizes for compact local representations on each device alongside a global model spanning all devices. Collins et al. (2021) introduced FedRep, which learns a shared data representation among clients while maintaining unique

local heads to enhance each client's model quality. Model Contrastive Learning (MOON), presented by Li et al. (2021), improves local update consistency by maximizing alignment between representations learned from local and global models. Additionally, Tan et al. (2022) introduces a novel Federated Prototype-wise Contrastive Learning (FedPCL) approach that uses pre-trained neural networks as backbones, facilitating knowledge sharing through class prototypes and building client-specific representations via prototype-wise contrastive learning. FedCA, proposed by Zhang et al. (2023), aggregates representations from each client, aligning them with a base model trained on public data to mitigate inconsistencies and misalignment in the representation space across clients. TurboSVM-FL, introduced by Wang et al. (2024), accelerates convergence in federated classification tasks by employing support vector machines for selective aggregation and applying max-margin spread-out regularization on class embeddings. Despite these advancements, research in federated representation learning specific to reinforcement learning remains limited.

**Federated Reinforcement Learning**   Federated Reinforcement Learning enables clients to collaboratively learn a unified policy while preserving privacy by avoiding the exchange of raw trajectories. Notably, Fan et al. (2021) proposed Federated Policy Gradient with Byzantine Resilience (FedPG-BR), which addresses convergence and fault tolerance against adversarial attacks or random failures in homogeneous environments using variance-reduced policy gradient methods. However, it does not consider the challenges posed by heterogeneous environments, which is the focus of our work. To address environmental heterogeneity, Jin et al. (2022) introduced QAvg and PAvg algorithms, employing value function-based and policy gradient methods. They further proposed personalized policies that embed environment-specific state transitions into low-dimensional vectors, improving both generalization and training efficiency. Similarly, Tang et al. (2022) developed FeSAC, a method based on the soft actor-critic framework. FeSAC isolates local policies from global integration and employs trend models to adapt to regional disparities. Building on these advancements, our work focuses on learning federated behavioral metric-based state projection function to effectively generalize across diverse environments. This approach enhances both policy robustness and value function generalization. To clearly differentiate our contributions, we provide a detailed comparison in Appendix B, outlining the distinctions in objectives, methodologies, and heterogeneity-handling mechanisms between FedRAG and prior works. This highlights how FedRAG advances generalization capabilities and cross-environment adaptability beyond existing methods.

**Behavioral Metrics-based Representation Learning**   Behavioral metric-based representation learning aims to create an embedding space that preserves behavioral similarities based on transitions and immediate rewards. Ferns et al. (2011) proposes using bisimulation metrics to measure state behavioral similarities in probabilistic transition systems for continuous state-space Markov Decision Processes (MDPs). On-policy bisimulation metrics introduced by Castro (2020) focus on behaviors specific to a given policy $\pi$, incorporating a reward difference term and the Wasserstein distance between dynamics models. To address the computational challenges associated with the Wasserstein distance, the MICo distance proposed by Castro et al. (2021) was developed to compare dynamics model distributions by measuring the distance between sampled subsequent states. The Conservative State-Action Discrepancy presented by Liao et al. (2023) separates the learning of the RL policy from the metric itself, focusing on the most divergent reward outcomes between states taking the same actions to define similarity in the embedding space. Chen & Pan (2022) propose the Reducing Approximation Gap distance to recursively measure expected states over dynamics models, focusing on sampling from the policy $\pi$ rather than the dynamics models. This approach reduces approximation errors and is particularly effective for representation learning. In our work, we apply approximation behavior metric-based representation learning to develop local state projection functions, capturing task-relevant behavioral similarities within each client's environment. Federated Learning then allows for sharing the parameters of these local projection function, enabling clients to benefit from generalized state representations across diverse environments.

## 3   PRELIMINARIES

This section highlights the Federated Soft Actor-Critic (FeSAC) variant central to our research. Soft Actor-Critic (SAC) is an off-policy actor-critic algorithm based on the maximum entropy RL framework (Haarnoja et al., 2018a). It aims to maximize future cumulative rewards and maximum entropy to increase robustness and exploration capabilities while avoiding policy convergence to suboptimal solutions. FeSAC is a federated variant of SAC, designed to facilitate collaborative

training among clients distributed across diverse environments, while ensuring the privacy of their respective data. The global environment $E = \{E^1, E^2, \ldots, E^N\}$ is composed of $N$ distinct local environments, and each client $k$ operates within its own unique local environment $E^k$. The transition probabilities differ across local environments, i.e., $P(s_{t+1}^i|s_t^i, a) \neq P(s_{t+1}^j|s_t^j, a), i \neq j$.

As the primary focus of our study is to investigate the application of approximated behavioral metric-based representation learning in federated reinforcement learning, we introduce the state projection function when discussing FeSAC. In the scope of representation learning for deep RL, a state projection function $\phi_{\omega^k}$ maps a high-dimensional state to low-dimensional vector, from which the policy $\pi_{\psi^k}(a|\phi_{\omega^k}(s))$ is learned. We configure all critic networks, target critic networks, and action networks to take the state representation $\phi_{\omega^k}(s)$ as input instead of the raw state $s$.

Unlike traditional FRL, the objective of FeSAC is to derive a set of maximum entropy policies that are specifically optimized for their respective local environments. The target policy $\widetilde{\pi}^k$ for client $k$ in its local environment $E^k$ is as follows:

$$\widetilde{\pi}^k = \arg\max_{\pi^k} \sum_{t=0}^{T} \mathbb{E}_{(s_t^k, a_t^k) \sim \tau_{\pi^k}} \left[ \gamma^t r(s_t^k, a_t^k) + \alpha^k H(\pi^k(\cdot|\phi_{\omega^k}(s_t^k))) \right], \tag{1}$$

where $s_t^k$ and $a_t^k$ represent the state and action made by client $k$ in its local environment $E^k$ at time $t$; $\tau_{\pi^k}$ refers to the trajectory generated by the policy $\pi^k$ of client $k$, which encompasses the sequence of states and actions over time; $\gamma^k$ is the discount rate; $\alpha^k$ is the entropy regularization coefficient used to control the importance of entropy; $\mathcal{H}(\pi^k(\cdot|\phi_{\omega^k}(s_t^k))) = E[-log\pi^k(\cdot|\phi_{\omega^k}(s_t^k))]$ represents the entropy of the policy.

To evaluate the impact of the policy on local environments, the soft state value is defined as:

$$V(s_t^k) = \mathbb{E}_{a_t^k \sim \pi_{\psi^k}} \left[ Q_{\theta^k}(\phi_{\omega^k}(s_t^k), a_t^k) - \alpha^k \log \pi_{\psi^k}(a_t^k|\phi_{\omega^k}(s_t^k)) \right], \tag{2}$$

where $Q_{\theta^k}$ denote the local critic Q network for client $k$. Each client adjusts its local Q-network to approximate the global Q-network, thus leveraging global knowledge while retaining its own characteristics:

$$L_Q(\theta^k) = \mathbb{E}_{(s_t^k, a_t^k, r_t^k, s_{t+1}^k) \sim \mathcal{D}^k} \left[ \left( Q_{\theta^k}(\phi_{\omega^k}(s_t^k), a_t^k) - \left( r_t^k + \gamma V_{\bar{\theta}}(s_{t+1}^k) \right) \right)^2 \right], \tag{3}$$

where $V_{\bar{\theta}}$ denotes use the target critic Q networks to calculate the soft state value. In FeSAC, the target critic Q network refers to the global critic Q network, which is broadcasted by the server to all clients. The global critic Q network $Q_{\bar{\theta}}$ is formed by aggregating the local critic Q networks of each client through soft updates, considering the reward differences of state-action pairs in each client's environment to obtain a value estimation in a global context:

$$Q_{\bar{\theta}} \leftarrow \epsilon Q_{\theta^k} + (1 - \epsilon)Q_{\bar{\theta}}, \quad k \in \{1, 2, \ldots, N\}, \tag{4}$$

where $\epsilon$ is the aggregation factor.

The updated local Q-network then guides the update of the local policy, which keeps the local variability as well as learning the implicit trend of the global environment:

$$L_\pi(\psi^k) = \mathbb{E}_{s_t^k \sim \mathcal{D}^k} \left[ \mathbb{E}_{a_t^k \sim \pi_{\psi^k}(\cdot|\phi_{\omega^k}(s_t^k))} \left[ \alpha^k \log \pi_{\psi^k}(a_t^k|\phi_{\omega^k}(s_t^k)) - Q_{\theta^k}(\phi_{\omega^k}(s_t^k), a_t^k) \right] \right]. \tag{5}$$

The temperature parameter $\alpha^k$ is adapted to balance exploration and exploitation by controlling the relative importance of the entropy term in the policy's objective. The update objective for $\alpha^k$ in client $k$ is as follows (Haarnoja et al., 2018b):

$$L_\alpha(\alpha^k) = \mathbb{E}_{s_t^k \sim \mathcal{D}^k} \left[ \mathbb{E}_{a_t^k \sim \pi_{\psi^k}(\cdot|\phi_{\omega^k}(s_t^k))} [\alpha^k \log \pi_{\psi^k}(a_t^k|\phi_{\omega^k}(s_t^k)) - \alpha^k \bar{\mathcal{H}}] \right], \tag{6}$$

where $\bar{\mathcal{H}}$ is a target entropy level to tune the degree of exploration and $\bar{\mathcal{H}} = -|\mathcal{A}|$.

## 4 METHODOLOGY

In this section, we present the problem formulation for federated reinforcement learning with heterogeneous environments, introduce the approximated behavioral metric-based state projection function, propose the FedRAG framework and provide a theoretical analysis of its privacy preserving.

### 4.1 PROBLEM FORMULATION

In federated reinforcement learning with heterogeneous environments, $N$ clients each interact with their own unique local environment $E^k$, each modeled as a unique Markov Decision Process (MDP): $\{S^k, A, r^k, P^k, \gamma\}$. Each client has a unique state space $S^k$, reward function $r^k(s, a)$, and state transition dynamics $P^k(s'|s, a)$, reflecting the diversity of their environments, while sharing a common action space $A$ and discount factor $\gamma$. A central server facilitates collaboration by periodically aggregating and distributing shared model parameters, specifically the state projection function $\phi_\omega$ in FedRAG. This function maps local states to a shared embedding space, enabling clients to benefit from collective learning while preserving privacy. FedRAG optimizes local policies $\pi^k(s|a)$ by sharing a state projection function $\phi_\omega$, aiming to maximize cumulative reward and entropy:

$$\widetilde{\pi}^k = \arg\max_{\pi^k} \frac{1}{N} \sum_{i=1}^{n} \left\{ \sum_{t=0}^{\infty} \mathbb{E}_{(s_t^k, a_t^k) \sim \tau_{\pi^k}} \left[ \gamma^t R^k(s_t^k, a_t^k) + \alpha^k H(\pi^k(\cdot | \phi_{\omega^k}(s_t^k))) \right] \right\}, \quad (7)$$

where $a_t^k \sim \pi^k(\cdot | s_t^k)$, $s_{t+1}^k \sim P^k(\cdot | s_t^k, a_t^k)$ and $k \in \{1, 2, \ldots, N\}$. To preserve data privacy, only the parameters of the state projection function $\omega$ are shared between clients and the server. Raw states, rewards, and transition dynamics remain local to each client, ensuring that sensitive information is not exchanged while still enabling effective federated learning.

### 4.2 CLIENT RAG DISTANCE

In FeSAC, clients in different environments share knowledge by aligning their local Q networks with the global Q network. This enables them to learn optimal local policies while adapting to network changes. However, as environments become complex, clients may struggle to capture task-relevant information, as shown in Section 5.2. Consequently, the global perception after federation becomes unclear, hindering effective adaptation to environmental changes. To enhance generalization in complex environments, we introduce behavior metric-based representation learning into FeSAC. This approach learns robust state representations that filter out task-irrelevant background information, speeding up the learning process and improving policy generalization across diverse environments.

For each client $k$, behavioral metric-based representation learning is to learn a local state encoding network $\phi_{\omega^k} : S^k \to \mathbb{R}^n$ with parameters $\omega^k$, which can be cast as a minimization problem of the loss between the distance on the embedding space, $\hat{d}(\phi_{\omega^k}(s_i^k), \phi_{\omega^k}(s_j^k))$, and the corresponding behavior metric, $d^\pi(s_i^k, s_j^k)$, between any pair of states $s_i^k$ and $s_j^k$:

$$L_\phi(\omega^k) = \mathbb{E}\left[ \left( \hat{d}(\phi_{\omega^k}(s_i^k), \phi_{\omega^k}(s_j^k)) - d^\pi(s_i^k, s_j^k) \right)^2 \right]. \quad (8)$$

The Reducing Approximation Gap (RAG) distance is a behavioral metric that measures the absolute difference between the reward expectations of two states and the distance between the next state expectations of dynamics models. And it is defined as follows:

$$d^\pi(s_i^k, s_j^k) = \left| \mathbb{E}_{a_i^k \sim \pi^k} r_{a_i^k}^{s_i^k} - \mathbb{E}_{a_j^k \sim \pi^k} r_{a_j^k}^{s_j^k} \right| + \gamma \mathbb{E}_{a_i^k \sim \pi^k, a_j^k \sim \pi^k} d(\mathbb{E}[s_{i+1}^k], \mathbb{E}[s_{j+1}^k]), \quad (9)$$

where $\mathbb{E}_{a_i^k \sim \pi^k} r_{a_i^k}^{s_i^k}$ represents the expected reward obtained by taking action $a_i^k$ in state $s_i^k$ under the policy $\pi^k$ of client $k$, $\mathbb{E}[s_{i+1}^k] = \mathbb{E}_{s_{i+1}^k \sim P_{a_i^k}^{s_i^k}}[s_{i+1}^k]$ is the expectation value of next state over the dynamics model $P(s_i^k, a_i^k)$.

Then the approximation of RAG relax the computationally intractable reward difference term without introducing any approximate gap, as shown below:

$$d^\pi(s_i^k, s_j^k) = \sqrt{\mathbb{E}_{a_i^k \sim \pi^k, a_j^k \sim \pi^k} \left[ \left( r_{a_i^k}^{s_i^k} - r_{a_j^k}^{s_j^k} \right)^2 \right] - \mathrm{Var}[r_{s_i^k}] - \mathrm{Var}[r_{s_j^k}]}$$
$$+ \gamma \mathbb{E}_{a_i^k \sim \pi^k, a_j^k \sim \pi^k} d^\pi \left( \mathbb{E}_{s_{i+1}^k \sim P_{a_i^k}^{s_i^k}}[s_{i+1}^k], \mathbb{E}_{s_{j+1}^k \sim P_{a_j^k}^{s_j^k}}[s_{j+1}^k] \right). \quad (10)$$

For each client, because the reward variance $\mathrm{Var}[r_{s_i^k}]$ is computationally intractable, we can learn a neural network approximator to estimate it by assuming that the reward $r_{s^k}$ on state $s^k$ is Gaussian distributed. Let $R_{\xi^k}$ be the learned reward function approximation parameterized by $\xi^k$, which outputs a Gaussian distribution, $R_{\xi^k}(s^k) = \{\hat{\mu}(r_{s^k}), \hat{\sigma}(r_{s^k})\}$. These loss functions are as follows:

$$L_R(\xi^k) = \mathbb{E}_{(s^k, r^k) \sim \mathcal{D}^k} \left[ \frac{(r^k - \hat{\mu}(r_{s^k}))^2}{2\hat{\sigma}(r_{s^k})} \right], \tag{11}$$

where $\hat{\mu}$ and $\hat{\sigma}$ are the mean and the standard deviation, respectively.

Similarly, in order to estimate the expected next states $\mathbb{E}_{s_{i+1}^k \sim P_{a_i^k}^{s_i^k}} [s_{i+1}^k]$ for each client k, we learn a dynamics model $\hat{P}(\phi_{\omega^k}(s), a) = \{\hat{\mu}(\hat{P}_{\phi_{\omega^k}(s)}^a), \hat{\sigma}(\hat{P}_{\phi_{\omega^k}(s)}^a)\}$ for each client, which outputs a Gaussian distribution over the next state embedding:

$$L_{\hat{P}}(\eta^k) = \mathbb{E}_{(s,a,s') \sim D^k} \left[ \left( \frac{\phi_{\omega^k}(s') - \hat{\mu}(\hat{P}_{\phi_{\omega^k}(s)}^a)}{2\hat{\sigma}(\hat{P}_{\phi_{\omega^k}(s)}^a)} \right)^2 \right]. \tag{12}$$

Based on the above approximation, the RAG loss for each client can be defined as:

$$\begin{aligned}
L_{\mathrm{RAG}}(\phi_{\omega^k}) = \mathbb{E}_{D^k} \Big[ & \left( \hat{d}(\phi_{\omega^k}(s_i^k), \phi_{\omega^k}(s_j^k)) - \gamma \hat{d}(\hat{\mu}(\hat{P}_{\phi_{\omega^k}(s_i^k)}^{a_i^k}), \hat{\mu}(\hat{P}_{\phi_{\omega^k}(s_j^k)}^{a_j^k})) \right)^2 \\
& - \left( \left| r_{a_i^k}^{s_i^k} - r_{a_j^k}^{s_j^k} \right|^2 - (\hat{\sigma}(r_{s_i^k}))^2 - (\hat{\sigma}(r_{s_j^k}))^2 \right) \Big]^2,
\end{aligned} \tag{13}$$

where $D^k$ represents the replay buffer or the set of data collected from environment $k$ by the RL algorithm, e.g. SAC. Considering that the behavior metric has non-zero self-distance, the distance on the Embedding space adopts the approximate form proposed in MICo (Castro et al., 2021), which produces a non-zero self-distance and helps in maintaining proximity between similar states rather than pushing them apart:

$$\hat{d}(\phi(s_i^k), \phi(s_j^k)) = \|\phi(s_i^k)\|^2 + \|\phi(s_j^k)\|^2 + K\varphi(\phi(s_i^k), \phi(s_j^k)), \tag{14}$$

while $\varphi$ is absolute angle distance and $K$ is a hyper-parameter. The relevant properties and proofs of the RAG distance are displayed in Appendix C and Appendix D.

## 4.3 FedRAG Framework

Under the federated learning framework, we share the parameter $\omega$ of the state projection function $\phi_\omega$. The FedRAG framework operates with multiple clients and a federated central node. Each client $k$ generates local parameters $\omega^k$ for the state projection function and updates policy networks based on their local environment. The federated central node collects these local parameters $\omega^k$ from all clients, aggregates them into a global distribution, and then distributes the updated global parameters back to the clients. Specifically, each client uses the state projection $\phi_{\omega^k}(s)$ as input for both the actor and critic networks. We assume that global $\omega$ follows a Gaussian distribution, with each client learning only a portion of the overall distribution. Therefore, we add a Gaussian regularization term after the RAG regression function Eq. 13, leading to the new loss formulation:

$$\begin{aligned}
L_{\mathrm{FedRAG}}(\phi_{\omega^k}) = \mathbb{E}_{D^k} \Big[ & \left( \hat{d}(\phi_{\omega^k}(s_i^k), \phi_{\omega^k}(s_j^k)) - \gamma \hat{d}(\hat{\mu}(\hat{P}_{\phi_{\omega^k}(s_i^k)}^{a_i^k}), \hat{\mu}(\hat{P}_{\phi_{\omega^k}(s_j^k)}^{a_j^k})) \right)^2 \\
& - \left( \left| r_{a_i^k}^{s_i^k} - r_{a_j^k}^{s_j^k} \right|^2 - (\hat{\sigma}(r_{s_i^k}))^2 - (\hat{\sigma}(r_{s_j^k}))^2 \right) \Big]^2 + \frac{\lambda}{2} \|\omega^k - \omega^G\|_2^2,
\end{aligned} \tag{15}$$

where $\omega^G$ represents the expectation of the global Gaussian distribution.

Through the federated learning process, we upload $\omega^k$ to the server periodically. According to the central limit theorem, we approximate the global Gaussian distribution by summing the mean of all local $\omega^k$ at the server. Then server distributes result to each client, so that the local learning results

---

**Algorithm 1** FedRAG Algorithm

---

1: Initialize $\phi_{\omega^k} : S \to \Phi$, $\phi_{\bar{\omega}^k} : S \to \Phi$, $Q_{\theta^k} : \Phi \times A \to \mathbb{R}$, $Q_{\bar{\theta}^k} : \Phi \times A \to \mathbb{R}$, $\pi_{\psi^k} : \Phi \to$
$[0,1]^{|A|}$, $R_{\xi^k} : S \to \mathbb{R} \times \mathbb{R}_+$, $\hat{P}_{\eta^k} : \Phi \times A \to \mathbb{R}^{d_\Phi} \times \mathbb{R}_+^{d_\Phi}$, for $k \in \{1, 2, \ldots, N\}$.  ▷ Initialize local network parameters
2: Initialize $\phi_{\omega^G} : S \to \Phi$.  ▷ Initialize global network parameters at the federated center node
3: $\omega^k \leftarrow \omega^G$, $\bar{\omega}^k \leftarrow \omega^G$ for $k \in \{1, 2, \ldots, N\}$.  ▷ Equalize global state projection network parameters and local projection network parameters
4: $D^k \leftarrow \emptyset$ for $k \in \{1, 2, \ldots, N\}$.  ▷ Initialize an empty replay memory
5: **while** running **do**
6:   **for** each client $k \in \{1, 2, \ldots, N\}$ **do**
7:     Get state $s_t$ from the environment $E^k$
8:     $a_t \sim \pi(a_t | \phi_{\omega^k}(s_t))$.  ▷ Sample action from the client $k$
9:     $s_{t+1} \sim P(s_{t+1} | s_t, a_t)$.  ▷ Sample transition from the environment $E^k$
10:     $D^k \leftarrow D^k \cup \{(s_t, a_t, r(s_t, a_t), s_{t+1})\}$.  ▷ Store the transition in replay memory
11:     $\theta^k \leftarrow \theta^k - \lambda_Q \hat{\nabla}_{\theta^k} L_Q(\theta^k)$.  ▷ Update local Q networks using Eq.(3)
12:     $\psi^k \leftarrow \phi^k - \lambda_\pi \hat{\nabla}_{\psi^k} L_\pi(\psi^k)$.  ▷ Update policy networks using Eq.(5)
13:     $\alpha^k \leftarrow \alpha^k - \lambda_\alpha \hat{\nabla}_{\alpha^k} J(\alpha^k)$.  ▷ Update temperature using Eq.(6)
14:     $\eta^k \leftarrow \eta^k - \lambda_\eta \hat{\nabla}_{\eta^k} L_{\hat{P}}(\eta^k)$.  ▷ Update dynamics model using Eq.(12)
15:     $\xi^k \leftarrow \xi^k - \lambda_\xi \hat{\nabla}_{\xi^k} L_R(\xi^k)$.  ▷ Update reward function using Eq.(11)
16:     $\omega^k \leftarrow \omega^k - \lambda_\omega \hat{\nabla}_{\omega^k} L_\phi(\omega^k)$.  ▷ Update state projection network using Eq.(15)
17:     $\bar{\theta}^k \leftarrow \tau_Q \theta^k + (1 - \tau_Q)\bar{\theta}$  ▷ Softly update target Q network
18:     $\bar{\omega}^k \leftarrow \tau_\phi \omega^k + (1 - \tau_\phi)\bar{\omega}^k$  ▷ Softly update target state projection network
19:     **if** running $n$ iterations **then**
20:       Upload $\omega^k$ to federated center node
21:     **end if**
22:   **end for**
23:   **if** in federated center node **then**
24:     $\omega^G \leftarrow \frac{1}{N} \sum_{k=1}^{N} \omega^k$.  ▷ Update global state projection network
25:     $\omega^k \leftarrow \omega^G$, $\bar{\omega}^k \leftarrow \omega^G$  ▷ Send global state projection network to clients
26:   **end if**
27: **end while**

---

are closer to the global distribution. Each client can maintain its own local training advantages while incorporating the global nature, and perform better when dealing with data outside of its own.

The proposed FedRAG is detailed in Algorithm 1. Initially, each client synchronizes its local state projection network with the global state projection network and preserves a global backup. Concurrently, each client initializes its other local networks such as critic network, target critic network, actor network, predictive transition dynamics model and predictive reward function. Clients operate individually with an empty replay buffer, interacting with their environments, to collect states, actions, rewards and next states, which are stored in the buffer. Once the buffer reaches a set number of transitions, the main phase begins. During this phase, clients continue collecting data and update their local networks and temperature parameters $\alpha$ independently. After a specified number of local updates, each client $k$ uploads their local state projection function parameters $\omega^k$ to a federated central node. The central node aggregates these parameters to update the global state projection function parameters $\omega^G$, which are then distributed back to update each client's local parameters and global backups. This allows clients to enhance their local state projection function by incorporating insights gained from the global environment.

## 4.4 EFFECTIVENESS OF ANTI-ATTACK

One of the major issues in federated learning is preserving privacy. In our analysis, we consider the existence of semi-honest adversaries. The adversaries may launch privacy attacks to snoop on the training data of other participants by analyzing periodic updates (e.g., gradients) of the joint model during training (Zhu et al., 2019). Such kind of attacks is referred to as Bayesian inference attack (Zhang et al., 2022).

A Bayesian inference attack is an optimization process that aims to infer the private variable $D_k$ to best fit client k protected exposed information $W_k^S$ as

$$
\begin{aligned}
d^* &= \arg\max_d \log(f_{D_k|W_k^S}(d|w)) \\
&= \arg\max_d \log\left(\frac{f_{W_k^S|D_k}(w|d)f_{D_k}(d)}{f_{W_k^S}(w)}\right) \\
&= \arg\max_d [\log f_{W_k^S|D_k}(w|d) + \log f_{D_k(d)}]
\end{aligned}
\tag{16}
$$

where $f_{D_k|W_k^S}(d|w)$ is the posterior of $D_k$ given the protected variable $W_k^S$. According to Bayes's theorem, maximizing the log-posterior $f_{D_k|W_k^S}(d|w)$ on $D_k$ involves maximizing summation of $\log(f_{W_k^S|D_k}(d|w))$ and $\log(f_{D_k}(d))$. The former one aims to find $D_k$ to best match $W_k^S$, and the latter one aims to make the prior of $D_k$ more significant. The learned conditional distribution $f_{D_k|W_k^S}$ from the Bayesian inference attack reflects the dependency between $W_k^S$ and $D_k$, which determines the amount of information that adversaries may infer about $D_k$ after observing $W_k^S$. However, in our approach, the parameter $\omega$ that we participate in federated learning is related to the representation function $\phi$ of the state. From the loss $L_{FedRAG}(\phi_\omega)$ in Equation 15, we can also see that $\omega$ is only related to the mapped state and reward, and has nothing to do with our private data state. Therefore, our proposed FedRAG protects the privacy of local state information to a certain extent.

## 5 EXPERIMENT

### 5.1 EXPERIMENTAL SETTINGS

In this section, we evaluate the utility and generalization ability of FedRAG with DeepMind Control Suite (DMC). The DMC is a benchmark for control tasks in continuous action spaces with visual input (Tassa et al., 2018). We evaluate our method on several tasks, such as cartpole-swing, cheetah-run, finger-spin and walker-walk. As shown in Appendix A.5, we simulated different environments by altering key physical parameters for each task. We render 84×84 pixels and stack 3 frames as observation at each time step. As described in the previous section, each client projects state observation to embedding space by using local state projection network, and updates local SAC network for policy evaluation and improvement. Local state projection function is also updated by using the approximated behavioral metric.

To evaluate the effectiveness and generalization of our method, we firstly perform experiments on 2 settings: 1)**Local**: clients can only interact and update local network in their own different environments without information sharing; 2)**Federated**: clients interact with their respective environments, update local network with information sharing according to federated methods, and upload local information to the central server every 4 episodes.

In our study, we set an episode to consist of 125 environment steps, training over a total of 4000 episodes, which equates to 500,000 steps. For each setting, we evaluate the performance of each clients on all environments every 16 local update episodes. In the federated learning scenario, every 4 episodes, clients upload their local parameters, which the server then aggregates and redistributes as global parameters.

### 5.2 COMPARISON OF FEDRAG AND BASELINE PERFORMANCE

As illustrated in Figure 2, we compared our proposed FedRAG method ($\lambda = 0.001$) with FedAVG (equivalent to FedRAG with $\lambda = 0$), FeSAC, and Local methods in the CartPole task with varying pole lengths. We assessed the average episode reward and standard deviation achieved by the clients in other environments. The results show that clients in the Local group, trained exclusively in their own environment without federated learning, struggled to adapt to other environments, resulting in the lowest performance. FeSAC had limited effectiveness in capturing task-relevant information in complex states, leading to only modest performance improvements. In contrast, FedRAG outperformed FedAVG by effectively integrating the global state projection function during local updates, resulting in significant performance gains in other environments.

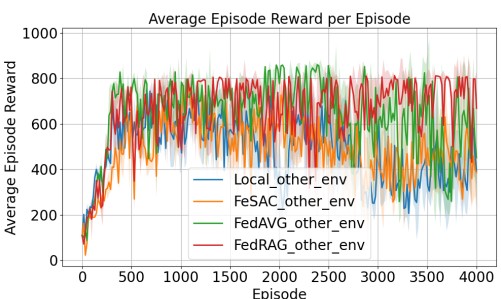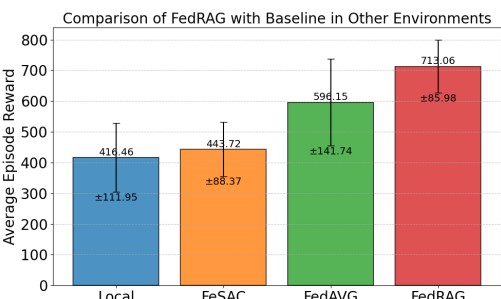

Figure 2: Comparison of FedRAG and Baseline in other environments.

## 5.3 TUNE THE PARAMETER $\lambda$

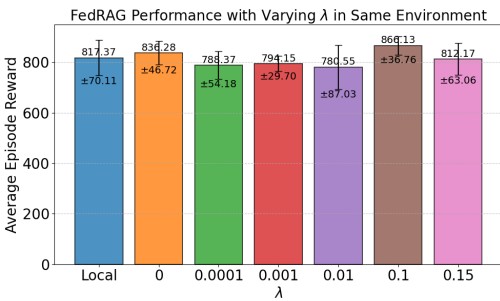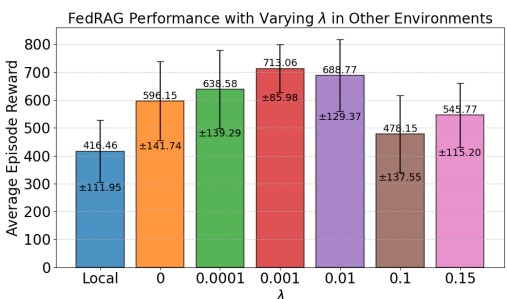

Figure 3: The results of varying lambda. In the left experiment, the training data and testing data are from environments with same setting, while in the right experiment, they are come from environments with different settings.

In the local update process of FedRAG, the regularization term in Equation 15 guides the local state projection function to align more closely with the global state projection function. We adjusted the regularization coefficient $\lambda$; a higher $\lambda$ enhances the consistency of local updates with the global network, while a lower $\lambda$ imposes fewer constraints. Setting $\lambda$ to zero simplifies the method to FedAvg. As shown in Figure 3, we compared the FedRAG method across various $\lambda$ values (0, 0.0001, 0.001, 0.01, 0.1, 0.15) with a non-federated approach to evaluate their impact on performance in both same and other environments. Increasing $\lambda$ enhances the effect of parameter sharing, clients obtain a global optimal state projection function more applicable to both the same environment and other environments, instead of focusing only on the same environment. In experiments focused on the same environment, both training and testing data came from same settings, revealing only minor fluctuations in performance among the federated methods, which underscores the robustness of our approach. For experiments involving other environments, increasing $\lambda$ enhanced the weight of the regularization term, allowing the locally learned state projection function to better align with the global state projection function and thus improving performance in other environments. The optimal performance was achieved at $\lambda = 0.001$. However, a large $\lambda$ may keep local updates too close to their initial global state, restricting parameter updates and slowing convergence. Overall, while performance remained stable in the same environment across all $\lambda$ values, notable improvements were observed in other environments, confirming the effectiveness of our federated approach.

## 5.4 PERFORMANCE IMPROVEMENT FOR FEDERATED LEARNING

In Figure 4, we compare the performance of the FedRAG method ($\lambda = 0.1/0.001$) with the Local approach by evaluating average episode rewards in both the same and other environments. The Local approach limits clients to their own environments, resulting in local optimal policies that poorly generalize. In contrast, FedRAG aggregates local state projection functions on a central server to create a global state projection function. By sharing this global function during local updates, clients benefit from cross-environment knowledge sharing while maintaining data privacy. With $\lambda = 0.1$,

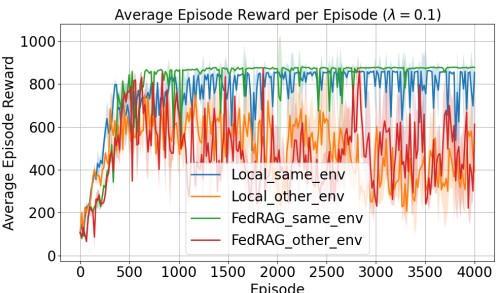 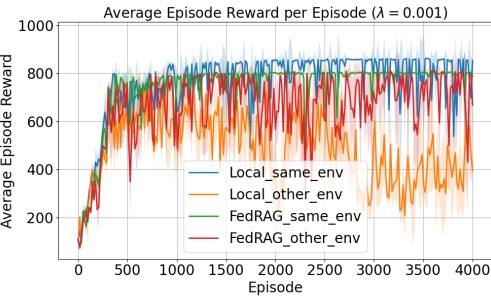

Figure 4: Comparison of Local and FedRAG with $\lambda = 0.1/0.001$ in same or other environments.

FedRAG enhances local performance by leveraging shared knowledge to overcome local optima, while also improving performance in other environments. At $\lambda = 0.001$, FedRAG achieves the best results in other environments with minimal loss in the same environment, demonstrating strong generalization and robustness across diverse settings.

## 5.5 FEDRAG PERFORMANCE ON VARIOUS DEEPMIND CONTROL TASKS

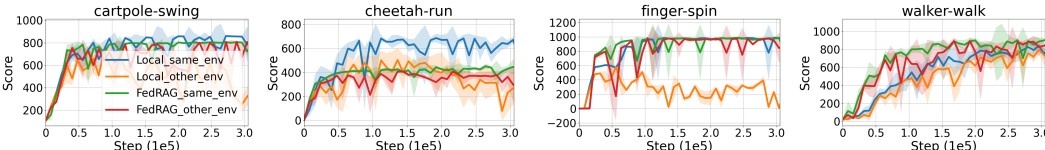

Figure 5: Experimental results on various DMC tasks.

To evaluate the robustness and effectiveness of our method, we conducted experiments on several tasks from DMC and compared the average episode rewards of clients using our FedRAG method with $\lambda = 0.001$ and the non-federated Local method in both same and other environments, as illustrated in Figure 5. In cartpole-swing and finger-spin tasks, FedRAG significantly outperformed the Local method in other environments while maintaining near-optimal performance in the same environment. This success stems from its federated approach, which integrates global knowledge while preserving local training advantages. In cheetah-run task, Local clients trained only on their own environments exhibited declining performance in other environments over time. In contrast, FedRAG maintained stable performance in other environments, benefiting from global knowledge. By the end of training, FedRAG outperformed the Local method in cross-environment evaluations. In walker-walk task, FedRAG demonstrated faster convergence and higher episode rewards across all environments, benefiting from federated state projection functions that enhanced task-relevant feature extraction and generalization. These results confirm the robustness and generalization of FedRAG across diverse tasks and environments.

The Appendix A presents additional experiments, including an ablation study on FedRAG components, evaluations under complex background distractions and generalization tests in unseen environments. These experiments demonstrate FedRAG's robustness, improved cross-environment adaptability, and strong generalization capability to new tasks.

## 6 CONCLUSION

Sharing the parameters of the approximated behavior metric-based state projection function enhances the performance of FRL and protects sensitive local information. In this work, we propose FedRAG, a FRL framework that shares the parameters of the state projections among clients. Under the FedRAG framework, we introduce a behavioral metric-based state projection function and develop its practical approximation algorithm in Federated Learning settings. We conduct empirical studies on several reinforcement learning tasks to verify the effectiveness of our proposed method.

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

## A EXPERIMENTAL DETAILS

### A.1 Q NETWORKS AND HYPERPARAMETERS

Table 1: Networks hyperparameters

| Hyperparameter | Value |
|---|---|
| Episode length | 1000 |
| Training steps | 500,000 |
| Replay buffer capacity | 20,000 |
| Batch size | 128 |
| Discount factor $\gamma$ | 0.99 |
| Optimizer | Adam |
| Networks learning rate | $5 \times 10^{-4}$ |
| $\log \alpha$ learning rate | $1 \times 10^{-4}$ |
| $\tau_\phi$ | 0.05 |
| $\tau_Q$ | 0.01 |
| Target Q-network update frequency | 2 |
| Actor network update frequency | 2 |
| $\alpha_{RAP}$ | 0.5 |
| $\alpha_P$ | $1 \times 10^{-4}$ |
| Actor log std bound | [-10, 2] |
| Action repeat for cartpole/cheetah | 8/4 |
| Action repeat for finger and walker | 2 |

Each client's Q networks include a state encoder $\phi_\omega$, which consists of stacked convolutional layers and a fully connected layer. It processes 3 stacked frames to produce the state representation $\phi_\omega(s)$ with input dimensions of $9 \times 84 \times 84$, convolutional kernels $[3, 3, 3, 3]$, 32 channels, and strides $[2, 1, 1, 1]$, resulting in an output dimension of 100. The Q-network has three fully connected layers with 1024 hidden units, taking input from $\phi_\omega(s)$ and action $a$. The actor network also consists of three fully connected layers that output the policy $\pi$. Both the dynamics model $\hat{P}$ and the reward function $R_\xi$ are two-layer MLPs with 512 hidden units, using ReLU activation. This architecture efficiently generates policies and Q-values from state inputs. Other hyperparameters are listed in Table 1.

### A.2 ABLATION STUDY ON FEDRAG CLIENT UPDATES

The FedRAG client update formula in Equation 15 has two key components for data sharing: replacing local parameters with global ones during distribution and applying L2 regularization to align local updates with global parameters.

To evaluate the impact of these components, we conducted ablation experiments, as shown in Figure 6. We compared four approaches: Local (no federated learning), Only_Replace (global parameters replace local ones without L2 regularization), Only_L2 (L2 regularization without replacing local parameters), and FedRAG (both global replacement and L2 regularization). The metrics measured were the average episode reward and standard deviation in different environments.

The results show that replacing local parameters with global ones improves generalization by leveraging shared knowledge, while L2 regularization enhances robustness by preventing overfitting. Omitting either component resulted in significant performance declines, confirming their essential role in our federated learning approach.

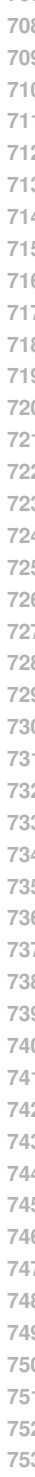

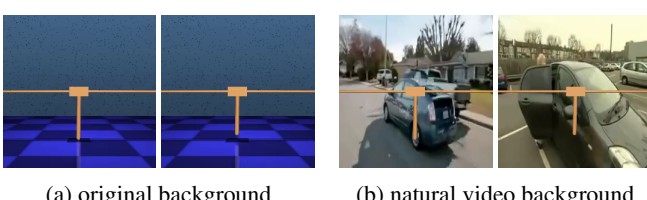

Figure 6: Performance comparison of FedRAG and variants in other environments.

### A.3 DISTRACTING DEEPMIND CONTROL SUITE

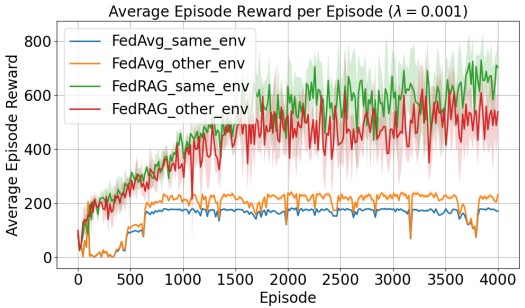

(a) original background        (b) natural video background

Figure 7: Illustrations of observations in DMC cartpole-swingup task for pole lengths 1.0 and 0.9.

Figure 8: Performance comparison of FedRAG and FedAVG with background distraction.

To evaluate the generalization and robustness of our method, we conducted experiments using the CartPole task in the DeepMind Control Suite, with background distractions and varying pole lengths to simulate different environments, as shown in Figure 7. We replaced the background with clips from the Kinetics dataset, which serves as a distraction for the RL algorithm. We selected 1,000 continuous frames from the video dataset for training the reinforcement learning clients and evaluated them using another 1,000 frames.

The results presented in Figure 8 demonstrate that FedRAG outperforms FedAVG in both the same and other environments, confirming that our method is more effective at learning generalizable state representations and better at capturing task-relevant information, even in complex settings.

### A.4 GENERALIZATION EVALUATION IN UNSEEN ENVIRONMENTS

To further evaluate the generalization ability of our proposed FedRAG method, we took the clients trained in Appendix 5.2 and tested them in a completely unseen environment, where none of the clients had prior exposure. We assessed their average episode reward, and the results are shown in Figure 9. Our method outperformed FeAVG and Local, achieving performance close to that of the client trained directly in the unseen environment. In contrast, FedSAC methods demonstrated

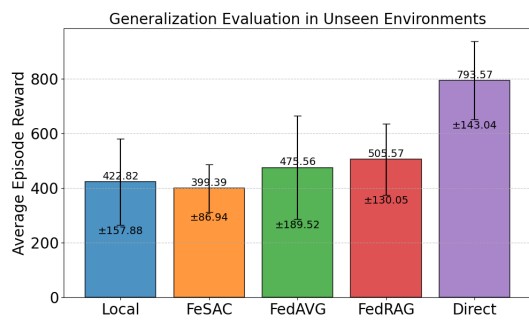

Figure 9: Comparison of FedRAG and Baseline in unseen environment.

poor performance, indicating that our approach enables clients to generalize more effectively to new, previously unseen environments.

### A.5 ILLUSTRATIONS OF OBSERVATIONS IN VARIOUS DMC TASKS

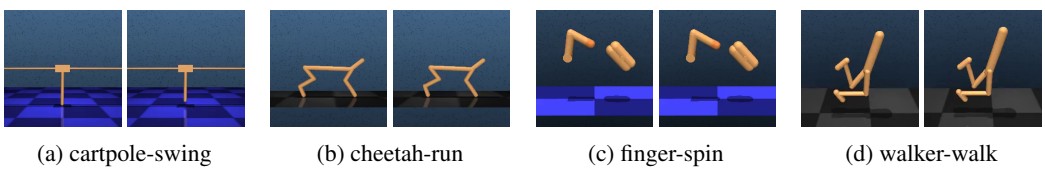

(a) cartpole-swing      (b) cheetah-run      (c) finger-spin      (d) walker-walk

Figure 10: Illustrations of observations in DMC tasks for pole lengths (1.0 and 0.9), cheetah torso lengths (1.0 and 0.9), finger distal lengths (0.16 and 0.18), and walker torso lengths (0.3 and 0.35)

As shown in Figure 10, we simulated different environments by modifying key physical parameters for several tasks from the DeepMind Control Suite, including cartpole-swing, cheetah-run, finger-spin, and walker-walk. Each task has a unique goal: balancing a swinging pole in cartpole-swing, maximizing speed in cheetah-run, rotating a finger in finger-spin, and simulating bipedal locomotion in walker-walk.

## B COMPARISON WITH RELATED WORK

To better position our work, we provide a detailed comparison with Fan et al. (2021) and Jin et al. (2022), highlighting the differences in objectives, methodologies, and contributions.

### B.1 COMPARISON WITH FAN ET AL. (2021)

**Objective:** Fan et al. (2021) proposed Federated Policy Gradient with Byzantine Resilience (FedPG-BR) to address convergence guarantees and fault tolerance in homogeneous FRL settings. Their focus is on filtering adversarial gradients and ensuring system robustness against Byzantine agents.

**Methodology:** The framework employs a variance-reduced federated policy gradient method. The server aggregates gradients sent by clients, applies a two-step Byzantine filtering rule, and updates the global policy. Clients compute gradients directly from their local trajectories without performing local updates.

**Limitations:** Fan et al.'s method assumes homogeneous environments and does not address heterogeneity among clients. It is tailored for variance-reduced policy gradients and lacks personalization mechanisms.

**Our Differences:**

- *Objective:* Unlike Fan et al., our work focuses on generalization across heterogeneous environments, enabling shared state projection functions to adapt to diverse dynamics.

Table 2: Comparison of FedRAG with related works

| Aspect | Fan et al. (2021) | Jin et al. (2022) | FedRAG (Our Work) |
|---|---|---|---|
| **Objective** | Convergence guarantees and fault tolerance in homogeneous environments. | Optimizing global policy across heterogeneous environments with personalization. | Generalization across heterogeneous environments via behavioral metric-based state representations. |
| **Methodology** | Variance-reduced federated policy gradient with Byzantine filtering. | Federated Q-network (QAvg) and policy network (PAvg) averaging, with environment embeddings. | Federating state projection parameters and updating local models with regularization. |
| **Handling Heterogeneity** | Assumes homogeneous environments. | Addresses heterogeneity via averaging and embeddings. | Tackles heterogeneity through shared state projection functions and behavioral metrics. |
| **Personalization** | Not addressed. | Environment embedding-based personalization for local policies. | Implicit personalization through regularized state projection updates. |
| **Contributions** | Theoretical guarantees for Byzantine-resilient FRL. | Suboptimality analysis under heterogeneity; embedding-based generalization. | Enhances policy robustness and generalization with behavioral metric-driven representations. |

- *Methodology:* FedRAG federates state projection parameters rather than policy gradients. Clients update their Q-networks and policy networks locally, regularized by the L2 norm between local and global parameters.

- *Others:* Instead of addressing Byzantine faults, our work tackles the challenges of heterogeneity and semi-honest adversaries, ensuring privacy and adaptability.

### B.2 COMPARISON WITH JIN ET AL. (2022)

**Objective:** Jin et al. (2022) tackled environmental heterogeneity by optimizing a global Q or policy while enabling personalization. They proposed QAvg and PAvg algorithms, along with a heuristic embedding-based personalization method.

**Methodology:** In QAvg and PAvg, agents perform local updates on Q or policy networks and share these updates with the server for aggregation. For personalization, they introduced embedding layers to capture unique environmental characteristics, enabling generalization to unseen environments through few-shot learning.

**Limitations:** While effective, Jin et al.'s approach relies heavily on averaging Q or policy parameters, which may not generalize well to environments with high variability.

**Our Differences:**

- *Objective:* While Jin et al. aim to optimize Q or policy networks, our focus is on behavioral metric-based state projection functions that enhance policy robustness across diverse environments.

- *Methodology:* Instead of federating Q or policy parameters, FedRAG aggregates state projection parameters, reducing sensitivity to environment-specific noise and improving cross-environment adaptability.

- *Others:* FedRAG's approach directly mitigates heterogeneity through shared projection functions, ensuring both generalization and robustness.

### B.3 NOVELTY OF FEDRAG

FedRAG introduces a novel perspective on federated reinforcement learning by:

- Developing approximated behavioral metric-based state projection functions for generalization across heterogeneous environments.
- Federating projection parameters to reduce communication overhead and enhance scalability.
- Balancing global consistency and local adaptability through regularized updates, enabling robust performance even in highly diverse settings.

These innovations bridge gaps in prior work, advancing the field of federated reinforcement learning.

## C  PROOF OF EQUATION 10

*Proof.* We first analyze the difference between $\mathbb{E}_{a_i^k \sim \pi^k, a_j^k \sim \pi^k} \left[ \left| r_{a_i^k}^{s_i^k} - r_{a_j^k}^{s_j^k} \right|^2 \right]$

and $\left| \mathbb{E}_{a_i^k \sim \pi^k} r_{a_i^k}^{s_i^k} - \mathbb{E}_{a_j^k \sim \pi^k} r_{a_j^k}^{s_j^k} \right|^2$ . The difference is given by:

$$\mathbb{E}_{a_i^k \sim \pi^k, a_j^k \sim \pi^k} \left[ \left| r_{a_i^k}^{s_i^k} - r_{a_j^k}^{s_j^k} \right|^2 \right] - \left| \mathbb{E}_{a_i^k \sim \pi^k} r_{a_i^k}^{s_i^k} - \mathbb{E}_{a_j^k \sim \pi^k} r_{a_j^k}^{s_j^k} \right|^2$$

$$= \mathbb{E}_{a_i^k \sim \pi^k} \left[ \left( r_{a_i^k}^{s_i^k} \right)^2 \right] + \mathbb{E}_{a_j^k \sim \pi^k} \left[ \left( r_{a_j^k}^{s_j^k} \right)^2 \right] - 2 \mathbb{E}_{a_i^k \sim \pi^k} \mathbb{E}_{a_j^k \sim \pi^k} \left[ r_{a_i^k}^{s_i^k} r_{a_j^k}^{s_j^k} \right]$$

$$- \left[ \mathbb{E}_{a_i^k \sim \pi^k} r_{a_i^k}^{s_i^k} \right]^2 - \left[ \mathbb{E}_{a_j^k \sim \pi^k} r_{a_j^k}^{s_j^k} \right]^2 + 2 \left[ \mathbb{E}_{a_i^k \sim \pi^k} r_{a_i^k}^{s_i^k} \right] \left[ \mathbb{E}_{a_j^k \sim \pi^k} r_{a_j^k}^{s_j^k} \right]$$

$$= \mathbb{E}_{a_i^k \sim \pi^k} \left[ \left( r_{a_i^k}^{s_i^k} \right)^2 \right] - \left[ \mathbb{E}_{a_i^k \sim \pi^k} r_{a_i^k}^{s_i^k} \right]^2 + \mathbb{E}_{a_j^k \sim \pi^k} \left[ \left( r_{a_j^k}^{s_j^k} \right)^2 \right] - \left[ \mathbb{E}_{a_j^k \sim \pi^k} r_{a_j^k}^{s_j^k} \right]^2$$

$$- 2 \mathbb{E}_{a_i^k \sim \pi^k, a_j^k \sim \pi^k} \left[ r_{a_i^k}^{s_i^k} r_{a_j^k}^{s_j^k} \right] + 2 \left[ \mathbb{E}_{a_i^k \sim \pi^k} r_{a_i^k}^{s_i^k} \right] \left[ \mathbb{E}_{a_j^k \sim \pi^k} r_{a_j^k}^{s_j^k} \right]$$

$$= \text{Var}[r_{s_i^k}] + \text{Var}[r_{s_j^k}] - 2\text{Cov}[r_{s_i^k}, r_{s_j^k}].$$

Since $r_{s_i^k}$ and $r_{s_j^k}$ are independent, $\text{Cov}[r_{s_i^k}, r_{s_j^k}] = 0$. Therefore, we have the reward difference term:

$$\left| \mathbb{E}_{a_i^k \sim \pi^k} r_{a_i^k}^{s_i^k} - \mathbb{E}_{a_j^k \sim \pi^k} r_{a_j^k}^{s_j^k} \right| = \sqrt{ \mathbb{E}_{a_i^k \sim \pi^k, a_j^k \sim \pi^k} \left[ \left| r_{a_i^k}^{s_i^k} - r_{a_j^k}^{s_j^k} \right|^2 \right] - \text{Var}[r_{s_i^k}] - \text{Var}[r_{s_j^k}]}.$$

$\square$

## D  PROPERTIES AND PROOFS OF THE RAG DISTANCE

**Theorem 1.** $d^\pi$ *is a contraction mapping w.r.t. the $L_\infty$ norm and has a unique fixed-point $D^\pi$.*

*Proof.* Let $D, D' \in \mathbb{M}$. We have

$$|d^\pi(D)(s_i, s_j) - d^\pi(D')(s_i, s_j)| = \left| \gamma \sum_{a_i, a_j} \pi(a_i|s_i)\pi(a_j|s_j)(D - D')(\mathbb{E}[s_i'], \mathbb{E}[s_j']) \right| \leq \gamma ||D - D'||_\infty.$$

Therefore, $d^\pi$ is a contraction mapping w.r.t. the $L_\infty$ norm and there exists a unique fixed-point for $d^\pi$ due to Banach's fixed-point theorem. This completes the proof.

Theorem 1 provides a convergence guarantee for the RAG distance that by iterating $d^\pi$, distance $D$ will converge to the fixed-point $D^\pi$. $\qquad\square$

**Theorem 2** (Value function difference bound). *Given states $s_i$ and state $s_j$, and a policy $\pi$, we have*

$$|V^\pi(s_i) - V^\pi(s_j)| \le D^\pi(s_i, s_j).$$

Theorem 2 demonstrates that the RAG distance between states upper-bounds the difference of their states values.

