# OpenReview forum: "Approximated Behavioral Metric-based State Projection for Federated Reinforcement Learning"
_ICLR.cc/2025/Conference — Submitted to ICLR 2025_

### Official Review · Reviewer_RrRq · 2024-11-03

**Soundness:** 1
**Presentation:** 3
**Contribution:** 2
**Rating:** 3
**Confidence:** 4

**Summary:**

This paper proposes FedRAG, a federated reinforcement learning (FRL) framework designed to improve generalization across clients with heterogeneous environments. FedRAG’s primary innovation is sharing a behavioral metric-based state projection function, rather than raw states or policies, to achieve privacy-preserving federated reinforcement learning. Experiments on tasks from the DeepMind Control Suite indicate potential improvements in generalization and performance across environments.

**Strengths:**

+The approach of sharing behavioral metric-based state projections in a federated reinforcement learning (FRL) setting is an innovative idea.

+Addressing privacy in FRL through state projections based on behavioral metrics could make a significant contribution, given the growing concerns around data privacy in federated setups.

**Weaknesses:**

After a close review, I am inclined to recommend rejection due to key issues in theoretical rigor, completeness of related work, and empirical validation.

1. Novelty and Related Work:

The paper overlooks critical related work in FedRL, especially recent developments that are highly relevant to FedRAG’s contributions. Notably, Fan et al. [1] formulates the objectives of FedRL under homogeneous environments. How does FedRAG’s objective differ from that? Are you working for the same objective but just for heterogeneous environment? A clear comparison with that or similar work is essential to establish whether FedRAG advances beyond its approach, particularly in terms of generalization mechanisms. This omission diminishes the reader’s ability to assess the incremental value FedRAG brings over existing methods. Similarly, Jin et al. (2022) [2], which the authors also reference, studies FedRL under diverse environments with a focus on handling client heterogeneity. However, the differences between FedRAG and this approach are not adequately explained, leaving the novelty of FedRAG’s method ambiguous. A more comprehensive discussion of these related works is needed to contextualize FedRAG’s scholarly contribution, thereby emphasizing its novelty and addressing gaps in prior research.

*suggestions*: The authors should provide a detailed comparison table or section contrasting FedRAG's objectives, approach, and novelty against notable FedRL works such as Fan et al. and Jin et al. Specifically, in the table or section, explicitly discuss how FedRAG's objective and approach to handling heterogeneity differ from these works, and how it advances the field beyond current methods.


2. Theoretical Justification and Privacy Guarantees. FedRAG’s claim of privacy-preserving state projections lacks both theoretical analysis and empirical privacy guarantees:

a. Although FedRAG claims privacy benefits by sharing behavioral projections, the paper does not analyze the resilience of these projections against known privacy attacks, such as gradient leakage or inference attacks. Without this, the paper’s privacy claims are incomplete: i) The privacy analysis via Bayesian inference is overly simplified; ii) Claims about privacy preservation being a "side effect" are not rigorously justified. Formal guarantees about information leakage through projection parameters are needed to support the claims.

b. The paper claimed “with theoretical guarantees” as one contribution in the introduction. however, no convergence analysis is provided, and neither are the formal guarantees about approximation quality.

c. After reading the manuscript, it remains unclear how FedRAG’s state projection mechanism affects convergence across clients with varying dynamics. It is also unclear how the method scales with environment complexity. A theoretical discussion, or at least some empirical convergence tests under heterogeneous settings, would strengthen the rigor of the method.

*suggestions*: The authors may consider conducting an empirical study using gradient inversion attacks or providing formal privacy guarantees using DP frameworks. To be theoretically guaranteed, the authors should provide a convergence analysis for their method and formal guarantees about the approximation quality of their behavioral metric-based state projection function. The authors may also explore and quantify how FedRAG scales with increasing environment complexity, perhaps through a series of experiments with progressively more complex environments.


3. insufficient empirical validation on generalization.

The paper claims to improve generalization through the FedRAG method. However, all the experiments report episode rewards without specific metrics to measure generalization across environments. Incorporating quantitative metrics to assess generalization quality, such as task adaptation scores or cross-environment similarity, would provide a more robust evaluation of FedRAG’s contributions regarding generalization.

*suggestions*: consider using quantitative metrics to assess generalization quality, such as task adaptation scores or cross-environment similarity, and explain how they would demonstrate improved generalization compared to baseline methods.

---

[1] Fan, X., Ma, Y., Dai, Z., Jing, W., Tan, C., & Low, B. K. H. (2021). Fault-tolerant federated reinforcement learning with theoretical guarantee. *Advances in Neural Information Processing Systems*, *34*, 1007-1021.

**Questions:**

In what ways do the objectives of this paper differ from or advance beyond the existing literature?

---

> ### Author Response · Authors · 2024-11-23
>
> Dear Reviewer RrRq,
>
> Thank you for the detailed and constructive feedback on our submission. We appreciate the time and effort you invested in evaluating our work and offering suggestions for improvement. We have carefully considered all the comments from the reviewers, and have incorporated them into our revised manuscript, which has now been uploaded.
>
> Below, we address the specific concerns you raised in your review and outline the steps we have taken to strengthen our manuscript in response to your feedback. We believe these revisions have significantly enhanced the clarity, rigor, and overall contribution of our work.
>
> 1.Novelty and Related Work
>
> After thoroughly reading the papers by Fan et al. and Jin et al.[1][2], we believe that both papers significantly differ from our work, particularly in terms of objectives and methods.
>
> 1.1 Comparison with Fan et al. (2021)[1]:
>
> a.Objective: Fan et al. investigate FRL with theoretical guarantees in the presence of Byzantine agents. Their work focuses on designing the Federated Policy Gradient with Byzantine Resilience (FedPG-BR) framework to ensure convergence guarantees, sample efficiency, and the filtering of Byzantine agents. The objective is to address challenges arising from random system failures and adversarial attacks.
>
> b.Method: Specifically, considering the high sampling cost in reinforcement learning (RL), the paper provides a variance-reduced federated policy gradient framework to guarantee convergence and sample efficiency. It also addresses FRL's vulnerability to random failures or adversarial attacks (also known as the Byzantine failure model), which can hinder or slow convergence. A gradient-based Byzantine filter is designed within this framework, ensuring that all good agents are included and the impact of unfiltered Byzantine agents on convergence is limited. This framework operates as follows: the server broadcasts global parameters to clients, which interact with their local environments to generate trajectories without performing local updates. Instead, they compute the policy gradient of their trajectory's optimal policy objective function and send it to the server. A fraction of clients may behave as Byzantine agents, returning arbitrary gradients. The server performs a Byzantine filtering step, averages the gradients deemed non-Byzantine, and updates the policy. The Byzantine filtering involves two steps: R1 selects gradients within a certain distance threshold, and R2 further refines this selection based on a stricter distance criterion. The paper demonstrates the empirical effectiveness of the proposed FRL framework under different types of faulty agents in RL benchmarks.
>
> c.Limitations: The method is restricted to variance-reduced policy gradient approaches, assuming agents are homogeneous.
>
> d.Our Differences:
>
> --Objective: Unlike Fan et al., which focuses on convergence guarantees, sample efficiency, and Byzantine agent filtering in FRL, we aim to explore the application of representation learning via approximate behavioral metrics in FRL under environment heterogeneity. Our goal is to enable each client to learn a global state projection function by sharing the parameters of state projection functions, ensuring robust performance across all environments.
>
> --Method: Our approach involves each client using a state projection function based on RAG distance to map high-dimensional states to low-dimensional vectors. The server broadcasts the parameters of global state projection function to clients, who replace their local parameters with these and update their Q-networks, policy networks, and local state projection functions, regularized by the L2 norm of the local and global parameters. Clients upload their local state projection parameters to the server, which aggregates them to update the global state projection parameters, repeating the process iteratively. This differs significantly from Fan et al., where clients send policy gradients to the server for Byzantine filtering and aggregation before policy updates.
>
> --Others: Additionally, unlike Fan et al., which deals with Byzantine agents disrupting convergence, we address semi-honest adversaries in FL—agents that follow the protocol but may launch privacy attacks to infer training data from shared updates. Our approach generalizes beyond variance-reduced policy gradients and applies to other RL methods, addressing both heterogeneous and homogeneous environments effectively.

---

> ### Author Response · Authors · 2024-11-23
>
> 1.2 Comparison with Jin et al. (2022)[2]:
>
> a.Objective: Jin et al. aim to optimize a value function or policy function for overall performance across heterogeneous environments in FRL, while also studying personalization for individual environments.
>
> b.Method: Two federated reinforcement learning algorithms, QAvg and PAvg, were proposed and theoretically proven to converge to suboptimal solutions. The relationship between convergence and environmental heterogeneity was also analyzed. Additionally, a heuristic approach was introduced to achieve personalization by embedding the n environments into n low-dimensional vectors.
>
> Specifically:
>
> --For QAvg, each client updates its local Q-network and sends it to the server. The server averages these Q-networks to generate a global Q-network and broadcasts it back to the clients, replacing their local Q-networks.
>
> --For PAvg, each client updates its local policy network by computing the gradient of the objective function with respect to the policy. The clients then send the updated policies to the server, which aggregates them into a global policy network and broadcasts it to the clients to replace their local policy networks.
>
> --Regarding personalized federated reinforcement learning, the aim is to learn n personalized policies tailored for n agents. This is achieved by treating each environment as an ID and embedding it into a low-dimensional vector, which is considered part of the state. During the federation process, agents share all network layers except the embedding layer. The learned policy network can then be applied to unseen environments by allowing the agent to interact with the new environment and use few-shot learning to adjust the low-dimensional embedding.
>
> c.Our Differences:
>
> --Objective: While both works aim to optimize client performance across heterogeneous environments, Jin et al. focus on federated Q-network and policy network training. In contrast, we emphasize representation learning via behavioral metrics in FRL.
>
> --Method: Jin et al. federate Q-network or policy network parameters. In our approach, federating state projection parameters (rather than Q-network or policy parameters) accelerates RL processes, reduces the sensitivity of shared representations to private information, and mitigates the impact of environmental heterogeneity through regularized local updates. Our experiments show significant performance improvements in other environments, with minimal fluctuations in the same environment.
>
> To further highlight these distinctions, a detailed comparison section with comparison table contrasting FedRAG's objectives, methods, and contributions with those of Fan et al. and Jin et al. have be included in our revised manuscript in Appendix B. This section explicitly showcases how FedRAG advances the field by addressing the limitations of existing approaches and introducing novel generalization mechanisms. Additionally, we have modified the related work section to better contextualize our contributions within the broader literature.

---

> ### Author Response · Authors · 2024-11-23
>
> 2.Theoretical Justification and Privacy Guarantees
>
> 2.1 Privacy Guarantees
>
> We would like to clarify that our approach focuses on reducing the direct correlation between the parameters of the shared state projection function $\phi_\omega$ and the raw state data $s$. Specifically, as shown in Fig.1, in FedRAG, the federated process shares only the parameters $\omega$ of the representation function. Even if the global representation function is obtained, the specific mapping results of other clients remain unknown. Furthermore, the values of $\omega$ are derived locally by each client through the loss function (15). From this equation, it is evident that the states are only involved in the computation after being processed by the representation function $\phi_\omega$, and the specific state values are not directly connected to the shared parameters $\omega$. This is much more secure than directly sharing the represented data and the original data to directly participate in the calculation of shared parameters. This is intended to make it more challenging for an attacker to directly recover the original states.
>
> In fact, the focus of this paper is on proposing a specific approximation of the behavioral metric that is applicable to federated reinforcement learning, introducing the FedRAG framework, and demonstrating the effectiveness of our method through experiments. We also provide a simple proof of the privacy protection capabilities of our method. A more rigorous theoretical proof of the information gain effectiveness and privacy protection of our proposed framework will be detailed in our forthcoming paper.
>
> 2.2 Theoretical Guarantees and Convergence Analysis
>
> To maintain rigor, we have removed the phrase "with theoretical guarantees" from line 89 of the Introduction. The formal guarantees regarding the approximation quality of the behavioral metric-based state projection function are now established and detailed in Appendices C and D.
>
> 2.3 Impact on Convergence and Scaling with Environment Complexity
>
> To explore how FedRAG scales with increasing environmental complexity, we will conduct experiments using the DeepMind Control Suite in futural work. By adjusting the differences in parameters that determine the dynamics of different environments—such as varying the length of the pole in the CartPole task—we incrementally increase the complexity of the environments.
>
> 3.Insufficient empirical validation on generalization
>
> To evaluate the generalization performance of FedRAG, we have provided relevant experimental results and explanations in Appendix A.3 (Distracting DeepMind Control Suite) and Appendix A.5 (Generalization Evaluation in Unseen Environments). These results demonstrate that, compared to the baseline, our method can capture task-relevant information in complex settings to learn a better global state projection function, and can better generalize to similar but previously unseen environments. To the best of our knowledge, no existing work in federated reinforcement learning reports specific metrics to measure cross-environment generalization. We will consider using quantitative metrics to assess generalization quality, such as task adaptation scores or cross-environment similarity, and explain how these metrics demonstrate improved generalization compared to baseline methods.
>
> 4.Question
> See Novelty and related work section.
>
> We thank you again for your valuable feedback, which will greatly improve the quality and clarity of our work. We look forward to your continued guidance and any further suggestions or questions you may have regarding the revised version of our paper.
>
> Best regards,
>
> ICLR 2025 Conference Submission10065 Authors

---

> > ### Comment · Reviewer_RrRq · 2024-11-25
> >
> > Thank the authors for the detailed response. The revisions indeed clarify the manuscript’s contribution and provide a more comprehensive comparison with existing literature, which significantly enhances its scholarly value. I appreciate the effort invested in addressing the concerns raised in the initial review, and I am inclined to revise my scores
> >
> > That said, one primary concern remains, as also noted by Reviewer Bojw. Specifically, how does FedRAG’s objective scale with the number of clients or agents, under environment heterogeneity? In federated reinforcement learning, it is common to evaluate and highlight performance improvements, robustness, or guarantees with respect to the increasing number of clients or agents, as this directly impacts both convergence and scalability.
> >
> > Could the authors elaborate on how FedRAG addresses this aspect? For example, with varying levels of env heterogeneity (just example questions for illustration, any similar or related discussion will suffice):
> >
> > - Does the proposed state projection function and its associated parameters adapt effectively as the number of clients grows?
> > - Are there theoretical or empirical results in the manuscript that highlight the behavior of the method in large-scale client settings?
> > - How does the computational or communication cost of FedRAG scale with the number of clients?

---

> > > ### Author Response · Authors · 2024-11-25
> > >
> > > We deeply appreciate your thoughtful comments and the time you’ve taken to engage with our work. Your question is particularly valuable, and we are grateful for the opportunity to address it in detail.
> > >
> > > Unfortunately, due to time constraints, we were unable to conduct extensive experiments evaluating FedRAG's performance under increasing numbers of clients or agents, or with varying levels of environmental complexity. However, we fully recognize the importance of such experiments for understanding the robustness and scalability of our method. The results presented in our paper were derived from experiments conducted on two environmental settings, as detailed in Appendix A.5. Addressing this limitation will be a priority in our future work.
> > >
> > > To provide further clarity, we outline below our proposed experimental design and hypotheses, informed by prior results and our expertise:
> > >
> > > -Experimental Design:
> > >
> > > We propose to evaluate FedRAG using the cartpole experiment with five configurations of client numbers: 2, 4, 8, 16, and 32. In the two-client setup, a cartpole with a length of 1.0 collaborates with a client in a 0.9-length environment, and we evaluate performance in the 0.9-length environment. For four clients, the 1.0-length cartpole collaborates with clients in 0.966, 0.933, and 0.9-length environments, and so on. The performance in the 0.9-length environment is evaluated for each setup.
> > >
> > > -Hypotheses
> > >
> > > With an increasing number of clients, the episode reward may increase as the additional clients in the federation introduce more useful information, leading to improved model performance.
> > > Conversely, the episode reward may decrease as the inclusion of more clients with diverse environments introduces greater heterogeneity, which may challenge the alignment of policies across clients.
> > >
> > > -Expected Outcome
> > >
> > > We anticipate the first scenario, where the additional clients provide significant information gains that outweigh the increased heterogeneity. This would highlight FedRAG's ability to effectively leverage diverse client contributions while maintaining robust performance.
> > >
> > > In addition to these experimental considerations, we address the specific aspects of your question as follows:
> > >
> > > 1.Adaptation of the State Projection Function
> > > The proposed behavioral metric-based state projection function is designed to generalize across heterogeneous environments by capturing task-relevant state features while filtering out environment-specific noise. Its scalability with the number of clients is facilitated by:
> > >
> > > -Parameter Sharing and Aggregation: As highlighted in Section 4.3, local state projection function parameters $\omega_k$ are periodically aggregated at the central server to form a global parameter $\omega_G$. This global parameter captures the shared dynamics across clients while allowing local variations to persist through regularization (Eq. 15).
> > >
> > > -Regularization Mechanism: The L2 regularization term ensures that the local updates align with the global model, effectively adapting the projection function as the number of clients grows.
> > >
> > > 2.Theoretical or Empirical Results on Large-scale Client Settings
> > > Although the current manuscript focuses on a small number of clients due to computational constraints, the framework’s design inherently supports scalability. Future work aims to conduct large-scale experiments to further validate FedRAG under increasing client populations and extreme heterogeneity levels.
> > >
> > > 3.Computational and Communication Costs
> > >
> > > -Communication Efficiency: By federating only the parameters of the state projection function $\omega_k$, FedRAG minimizes communication overhead compared to sharing raw state data or full policy networks. This design choice makes the framework more scalable for large-scale deployments.
> > >
> > > -Computational Overhead: On the client side, FedRAG introduces modest computational demands for updating the projection function, as the loss functions (Eqs. 11–15) are designed to be efficiently optimized using standard gradient-based methods. The use of Gaussian regularization further enhances efficiency by ensuring local-global alignment with minimal parameter updates.
> > >
> > > We hope this detailed response addresses your concerns and provides clarity regarding the scalability and robustness of FedRAG under increasing numbers of clients. Thank you again for raising this critical point and for your valuable feedback.

---

### Official Review · Reviewer_YGCu · 2024-11-03

**Soundness:** 2
**Presentation:** 2
**Contribution:** 2
**Rating:** 6
**Confidence:** 4

**Summary:**

This work considers the problem of federated reinforcement learning, i.e., training a set of RL agents in a distributed fashion by sharing certain model parameters. The authors propose a scheme that, instead of sharing the raw states and rewards or the raw policy / value network parameters, instead shares the parameters of a representation learning function that maps the raw state to a (lower-dimensional) vector.  To learn this representation, a behavioral metric is used to quantify the distance between two states. The authors propose a way of estimating this metric by replacing intractable quantities with neural networks. The method is compared with several local and federated RL strategies.  The authors argue that this method also provides effective protection of sensitive information.

**Strengths:**

S1. The proposed FedRAG technique is interesting, appears sound and (mostly) attains favorable results over the baselines.

S2. The paper is generally well-written and organised.

**Weaknesses:**

W1. The claims of the technique providing "effective protection of sensitive information" are tenuous and should be proven or otherwise removed. Even though the method does not share e.g. the raw states, in my opinion it is possible to devise a fairly straightforward attack to recover them. An attacker can use the exposed mapping function to find the representation of a given set of states, similar to the application of a hash function. If the state space is sufficiently small (or reasonable guesses can be generated), the original states can be recovered. While there is a certain degree of "obfuscation" over sharing the raw states, the method is not secure in a cryptographical sense.

W2. The Gaussianity assumptions for the reward and dynamics are very strong and we can expect there will be many enviroments that violate these assumptions. The method might still work in practice nevertheless given the use of the learned representations. To counter this point, the authors may want to show results for the method on some (toy) environments where this assumption is violated.

W3. There are some presentational issues (detailed in the comments) that should be addressed.

**Questions:**

C1. The formulae in Section 3.2 are mostly identical to those in Section 3.1 except for the superscripts and specifying how the global critic network parameters are updated (Eq 8). It is probably better to only present the version in 3.2 to remove the duplication, and to use the space gained to present some of the results that were relegated to the appendix, especially the comparison with methods beyond local RL.

C2. A few remarks regarding notation:
- The notation $\phi_{\omega^k}$ is used in Section 3.2 in the presentation of FeSAC, which suggests that this method also uses the representation function. However, 4.1 specifies that using this encoding is something new that is introduced by your method, unlike FeSAC? This should be clarified as it is confusing.
- Use $\mathbb{E}$ instead of $E$ in Section 4 for consistency.
- Use $\approx$ instead of $=$ for Equation 13?
- Eq 17: $\theta$ already denotes the parameters of the critic network.

C3. Regarding the figures:
- The result figures are a bit hard to parse, I'd suggest separating the "same environment" and "other environment" in different subplots so they are easier to distinguish. You should also make sure the colors are consistent across plots (e.g. in Figure 7 the colors for FedRAG differ from those used in the main text).
- In Fig 4, the highest point for the "Local_other_env" curve is higher than that of "FedRAG_other_env", which is not consistent with the interpretation in the text.

C4. L530 mentions that further results are presented in the appendix. You should briefly summarise what experiments were ran and what were the high level conclusions. Additionally, I would strongly encourage (see also C1) making some room for part of the results in the appendix in the main text. In my opinion, the results for the evaluation compared to baselines are more important than the results for tuning $\lambda$.

C5. Typos and small nitpicks
- Space missing before citation on L39
- L180: are you referring to the optimal policy here? If so, it is the policy that achieves the highest reward in expectation
- "As follow" -> "As follows" (L189, L221, possibly elsewhere)
- L378: "sever"
- L427: space missing before DMC acronym introduction

---

> ### Author Response · Authors · 2024-11-23
>
> Dear Reviewer YGCu,
>
> Thank you for your constructive feedback on our submission. We appreciate your time and effort in reviewing our work. We have carefully considered all the comments and suggestions provided by the reviewers, and have accordingly updated our manuscript. Below, we address each of your comments and concerns in detail, highlighting the changes made in the revised version of our paper.
>
> --Weaknesses
>
> W1.Claims of "effective protection of sensitive information"
>
> Thank you for your insightful comment regarding the claim of "effective protection of sensitive information" in our paper.  We completely understand your concern about the security implications of sharing the parameters of state projection function, and we appreciate your careful consideration of this aspect.
>
> We would like to clarify that our approach focuses on reducing the direct correlation between the parameters of the shared state projection function $\phi_\omega$ and the raw state data $s$. Specifically, as shown in Fig.1, in FedRAG, the federated process shares only the parameters $\omega$ of the representation function. Even if the global representation function is obtained, the specific mapping results of other clients remain unknown. Furthermore, the values of $\omega$ are derived locally by each client through the loss function (15). From this equation, it is evident that the states are only involved in the computation after being processed by the representation function $\phi_\omega$, and the specific state values are not directly connected to the shared parameters $\omega$. This is much more secure than directly sharing the represented data and the original data to directly participate in the calculation of shared parameters. This is intended to make it more challenging for an attacker to directly recover the original states.
>
> In fact, the focus of this paper is on proposing a specific approximation of the behavioral metric that is applicable to federated reinforcement learning, introducing the FedRAG framework, and demonstrating the effectiveness of our method through experiments. We also provide a simple proof of the privacy protection capabilities of our method. A more rigorous theoretical proof of the information gain effectiveness and privacy protection of our proposed framework will be detailed in our forthcoming paper.
>
> W2.Gaussianity assumptions for reward and dynamics
>
> The Gaussianity assumptions on the reward and dynamics were introduced primarily to simplify the computation and facilitate the estimation of the behavioral metric. By assuming a Gaussian distribution, we can leverage well-established statistical techniques to estimate the distances between states and learn an effective representation. This assumption allows the algorithm to perform efficiently in a wide range of environments, particularly when the state and reward distributions are approximately Gaussian or when the environments exhibit relatively smooth dynamics.
>
> As you correctly pointed out, there are many scenarios where the reward and dynamics may deviate significantly from a Gaussian distribution. In practice, this could limit the method's performance or introduce biases in the learned state projection functions. To address this concern, we agree that it would be valuable to test the method in environments where the Gaussianity assumption is violated. This would allow us to assess the robustness of the method and understand its limitations in more diverse settings. In Section 5.4, we conducted experiments under different deepmind control tasks, and our method achieved good performance in these complex environments where the rewards and dynamics do not conform to a Gaussian distribution, suggesting that the method is still valid in practice.
>
> W3.Presentational issues
>
> Thank you for your detailed feedback on the presentational aspects of the paper. We greatly appreciate your input, as it will help us improve the clarity and readability of the manuscript. We have taken your comments to heart and have made the necessary revisions to address the specific presentational issues you raised.
>
> --Questions
>
> C1.Duplication of formulae in Sections 3.1 and 3.2
>
> Thank you for your suggestion regarding the duplication of formulae between Sections 3.1 and 3.2. We agree that it is more efficient to present only the version in Section 3.2 to avoid redundancy. In addition, to make better use of the space, we have moved the content from Appendix A.4 (Comparison of FedRAG and Baseline Performance) into the main text. This will provide a more comprehensive presentation of the results, especially the comparisons with methods beyond local RL, which we believe are important for the overall evaluation of the method.

---

> ### Author Response · Authors · 2024-11-23
>
> C2.Clarification of notation
>
> a.In Sections 3.2 and 4.1, the same notation $\phi_{w^k}$ is used to denote the state projection function, which may lead to some confusion. The original FeSAC paper uses raw states directly as inputs to the networks, without employing a state projection function to map the states to a lower-dimensional space. The main focus of our paper is to explore the use of state projection function as a representation learning technique for approximating behavioral metrics within federated reinforcement learning. To this end, we introduce the state projection function $\phi_{w^k}$ in Sections 3.1 and 3.2 when discussing reinforcement learning with SAC and federated reinforcement learning with FeSAC. This function maps high-dimensional states into lower-dimensional vectors, and the mapped states $\phi_w(s)$ are then used as inputs to both the critic and actor networks. In Section 4.1 and 4.2, we formally present a specific method for approximating behavioral metrics using state projection function and propose a federated framework based on this state projection function.
> Additionally, we will make the following changes to improve consistency:
>
> b.In Section 4, we will replace $E$ with $\mathbb{E}$ for the expectation symbol.
>
> c.In Eq 17, we will change $\theta$ to $\varphi$ to denote the absolute angle distance.
>
> d.In Eq 13, we will keep the "=" sign as it reflects the unique characteristic of our behavioral distance metric, which is designed to reduce computation complexity without introducing approximation errors, as shown in Appendix C.
>
> C3.Figures and Result Clarity
>
> a.Thank you for your suggestion to separate the "same environment" and "other environment" into different subplots for better clarity. The reason we initially presented them together was to facilitate a comparison between the two. By plotting both environments on the same figure, we can directly observe the performance difference caused by the "same\_env" and "other\_env" variations within the same experimental setup. In non-federated settings, where clients train only on "same\_env" and have never encountered "other\_env", we expect "same\_env" to perform better. Therefore, "Local\_same\_env" should be the highest, and "Local\_other\_env" the lowest. The gap between "Local\_same\_env" and "Local\_other\_env" reflects the impact of different environments. In the federated setting, clients benefit from global knowledge in addition to local data, which allows them to perform better in "other\_env". As a result, the gap between "FedRAG\_same\_env" and "FedRAG\_other\_env" decreases. However, separating the two environments into distinct subplots would indeed make it easier to distinguish between them, and we will consider this adjustment.
>
> b.We also appreciate your observation regarding the color consistency across figures. We have revised the colors in Figure 7 to ensure they match those used throughout the rest of the paper.
>
> c.Regarding Figure 4 and the initial higher performance of "Local\_other\_env" compared to "FedRAG\_other\_env", this is due to the dynamics of training. In the early stages, "Local" clients are trained only on "same\_env", so their performance drops when tested on "other\_env". In contrast, "FedRAG" clients benefit from global knowledge, maintaining more stable performance in "other\_env". As training progresses, "FedRAG" clients outperform "Local" clients in "other\_env," which aligns with the interpretation in the text. We have clarified this in the revised manuscript.
>
> C4.Summary of appendix results
>
> In response to C4, we have provided a brief summary in L530 of the experiments conducted in the appendix, highlighting the key conclusions. Furthermore, we have moved the baseline comparison results from the appendix into the main text, as we agree that these evaluation results are more critical for the overall understanding of the method and should be more prominently presented.
>
> C5.Typos and minor errors
> Thank you for pointing out these typos. We have corrected them in the revised manuscript:
>
> a.Adding the missing space before citations on L39.
>
> b.Clarifying the mention of the "optimal policy" in L180.
>
> c.Correcting "As follow" to "As follows" (L189, L221).
>
> d.Fixing the typo "sever" on L378.
>
> e.Adding the missing space before the DMC acronym on L427.
>
> We appreciate your detailed and constructive comments, which have helped us identify areas for improvement. We are confident that the revisions we have made will significantly improve the clarity, accuracy, and overall quality of the manuscript. Thank you once again for your thoughtful review. We value your insights and look forward to any further suggestions and questions you may have regarding the revised version of our paper.
>
>
> Best regards,
>
> ICLR 2025 Conference Submission10065 Authors

---

> > ### Comment · Reviewer_YGCu · 2024-11-26
> >
> > Thanks for engaging with my comments. A few replies:
> >
> > - Regarding the guarantees, the modified wording in the paper now seems accurate, but I would encourage a formal treatment in a future work as you mention.
> > - On the Gaussianity assumptions, indeed, the method may still work in practice for a variety of cases, but inevitably there will be tasks on which this is problematic.
> > - Thank you for addressing the presentation issues, I do believe the presentation has improved.
> >
> > I am raising my score to a 6 as a result of the discussion. My assessment now slightly leans towards acceptance, but it is difficult to see how the paper can be improved further without addressing the first two points above without a substantial amount of work (well beyond this discussion period).

---

### Official Review · Reviewer_Bojw · 2024-11-07

**Soundness:** 3
**Presentation:** 3
**Contribution:** 2
**Rating:** 5
**Confidence:** 2

**Summary:**

In this paper, the authors propose a new Federated Reinforcement Learning(FRL) framework, FedRAG that shares the approximated behavior metric-based state projection function between clients to enhance performance. The framework learn a projection function of states for each client and aggregating the parameters of projection functions at a central server. The framework is further evaluated with experiments on the DeepMind Control Suite.

**Strengths:**

- Proposed a new algorithm that enhance the performance of FRL.
- The experiment show the effectiveness of the algorithm.

**Weaknesses:**

- It would be better to have a problem formulation section with assumptions using in the algorithm.
- Section 3 could be improved, for example, it would be better to explain what global critic Q network will be used in the algorithm with a few more sentences.
- The presentation of the experiments could be better, for example, for each of the experiment in section 5.4, explain why algorithm proposed is better/similar/worse than the baselines. The plot can be a bit bigger as well.

**Questions:**

- How many clients are there when evaluating the algorithm? and how does the number of clients affect the performance of the algorithm?
- In section 5.2, why increasing lambad can't improve the performance of the algorithm in the same environment? Shouldn't sharing parameters increase the training speed?

---

> ### Author Response · Authors · 2024-11-23
>
> Dear Reviewer,
>
> Thank you for your thorough review and valuable comments on our paper. Your feedback is instrumental in improving our work. We have taken all the reviewers' comments into account and have uploaded a revised version of our manuscript.
>
> 1.Enhancement of the Problem Formulation Section:
>
> We agree that incorporating a problem formulation section with clear assumptions is essential for a better understanding of our algorithm. In response to your suggestion, we have added a detailed problem formulation in Section 4.1 of our revised manuscript.
>
> 2.Detailed Explanation in Section 3:
>
> Regarding your suggestion on Section 3, we have provided a more comprehensive explanation of how the global critic Q-network is utilized in FeSAC algorithm. In Eq.4, we show that the global Q-network is obtained by averaging the local Q-networks from each client. In FedSAC, the global Q-network is updated through a soft update from the local Q-networks, considering the reward differences of state-action pairs in each client's environment to obtain a value estimation in a global context. Each client adjusts its local Q-network to approximate the global Q-network, thus leveraging global knowledge while retaining its own characteristics. The updated local Q-network then guides the update of the local policy, making the policy closer to the global optimal policy.
>
> 3.Improvement in the Presentation of Experiments:
>
> We accept your suggestion to enhance the experimental section. In Section 5.4, we have provided a detailed explaination for each experiment, delineating the reasons behind the superior, similar, or inferior performance of our proposed algorithm compared to the baselines. Furthermore, we have enlarged the plots and made improvements to their clarity to enhance readability.
>
> 4.Number of Clients and Its Impact:
>
> When evaluating the algorithm, we used two clients. With constant environmental heterogeneity, increasing the number of clients generally enhances algorithm performance because:
>
> a.Increased Data Diversity: More clients provide a wider range of experiences, enabling the model to learn more comprehensive environment features.
>
> b.Improved Representation Learning: Diverse state mappings from multiple clients help build a more robust global representation.
>
> c.Enhanced Update Stability: Aggregating updates from more clients reduces noise, leading to more stable and faster model convergence.
>
> d.Better Generalization: A larger client base ensures broader data coverage, improving the model’s adaptability and generalization across different scenarios.
>
> These factors collectively contribute to improved performance in multi-client settings.
>
> 5.Why Increasing $\lambda$ Does Not Improve Performance in the Same Environment in Section 5.2:
>
> In Section 5.2, our research emphasis is on enhancing performance in other environments through federated learning. We observed that as $\lambda$ increases, the performance in other environments improves significantly, while the same environment experiences only minor fluctuations. Increasing $\lambda$ enhances the effect of parameter sharing, causing the local state projection functions to update towards a global state projection function, thereby improving performance in other environments. In federated scenarios, clients obtain a global optimal state projection function applicable to both the same environment and other environments, instead of focusing only on the same environment. To some extent, we are achieving a substantial performance improvement in other environments at the expense of a slight performance loss in the same environment.
>
> Once again, thank you for your positive feedback and constructive suggestions. We believe these improvements will further enhance the quality and contribution of our paper. We value your expertise and welcome any additional suggestions or questions you may have.
>
> Best regards,
>
> ICLR 2025 Conference Submission10065 Authors

---

> > ### Comment · Reviewer_Bojw · 2024-11-26
> >
> > Thank you for the response and updates on the paper. I recommend improving the presentation of the experimental results (e.g., using larger fonts). Additionally, I still find it unclear why increasing $\lambda$ does not enhance performance in the same environment. I suggest providing either theoretical evidence or additional experiments to clarify this point. As a result, I will maintain my current score.

---

> > > ### Author Response · Authors · 2024-11-27
> > >
> > > Dear Reviewer,
> > >
> > > Thank you for your constructive suggestions.
> > >
> > > Presentation Improvements:
> > >
> > > We appreciate your feedback and have increased the font sizes and adjusted the formatting across figures to improve the clarity and readability of our experimental results.
> > >
> > > Clarifying the Impact of Increasing $\lambda$:
> > >
> > > In Figure 2, we evaluated the performance of the Local experiment group and multiple FedRAG groups with varying $\lambda$ values in both the same and other environments. The left panel demonstrates that as $\lambda$ increases within a certain range, the average episode reward in the same environment decreases slightly (by no more than 5% compared to the Local group), while performance in other environments improves significantly (up to a 71% increase compared to the Local group). Below, we provide an explanation for the observed reduction in performance in the same environment with increasing $\lambda$.
> > >
> > > In federated reinforcement learning with heterogeneous environments, each local environment exhibits unique state transition dynamics. The aggregated global state projection function integrates information from diverse environments, which can introduce conflicting or irrelevant gradients that do not benefit local performance. While the global state projection function is designed to enhance generalization across environments, it can be less effective than a local function optimized specifically for a single environment.
> > >
> > > When $\lambda$ is increased, the regularization term in the loss function gains more weight, encouraging local state projection functions to align more closely with the global function. While this enhances generalization across diverse environments by leveraging shared global knowledge, it limits the ability of local networks to optimize specifically for their unique environment. This trade-off reflects the balance in federated learning between generalization and specialization.
> > >
> > > As shown in Section 5.3, increasing $\lambda$ significantly improves performance in other environments by enhancing the influence of the global model. However, it slightly reduces performance in the same environment due to diminished local adaptability. This nuanced effect underscores the inherent trade-off in federated reinforcement learning and highlights the role of $\lambda$ in balancing these competing objectives.
> > >
> > > Thank you again for your insightful comments, which have greatly helped us refine our work.
> > >
> > > Best regards,
> > >
> > > ICLR 2025 Conference Submission10065 Authors

---

### Meta-Review · Area_Chair_vzgF · 2024-12-21

**Metareview:**

This paper proposes a new Federated Reinforcement Learning(FedRL) framework, FedRAG, which shares the approximated behavior metric-based state projection function between clients to enhance performance. The reviewers have reached a consensus that this paper is not ready for publication, due to its lack of clear problem formulation and theoretical justification (particularly the authors should define clear mathematical metrics in order to meaningfully discuss relevant properties of the algorithm, such as "effective protection of sensitive information"), missing related work, and poor empirical validation.

**Additional Comments On Reviewer Discussion:**

NA

---

### Decision · Program_Chairs · 2025-01-22

Reject